

# Status and prospects of SABRE North

Ambra Mariani[1,2,3★] on behalf of the SABRE North Collaboration

**1** Princeton University, Princeton, NJ 08544, USA
**2** INFN - Laboratori Nazionali del Gran Sasso, Assergi I-67100, Italy
**3** INFN - Sezione di Roma, Roma I-00185, Italy

★ ambra.mariani@roma1.infn.it

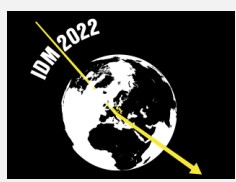

## Abstract

We present the characterization of a low background NaI(Tl) crystal for the SABRE North experiment. The crystal NaI-33, was studied in two different setups at Laboratori Nazionali del Gran Sasso, Italy. The Proof-of-Principle (PoP) detector was equipped with a liquid scintillator veto and collected data for about one month (90 kg×days). The PoP-dry setup consisted of NaI-33 in a purely passive shielding and collected data for almost one year (891 kg×days). The average background in the energy region of interest (1-6 keV) for dark matter search was 1.20 ± 0.05 and 1.39 ± 0.02 counts/day/kg/keV within the PoP and the PoP-dry setup, respectively. This result opens to a new shielding design for the physics phase of the SABRE North detector, that does not foresee the use of an organic liquid scintillator external veto, in compliance with the new safety and environmental requirements of Laboratori Nazionali del Gran Sasso.

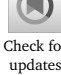

## 1 Introduction

The SABRE (Sodium-iodide with Active Background REjection) project plans to operate ultra-low background NaI(Tl) scintillating detectors to carry out a model-independent search of dark matter (DM) via the annual modulation signature, with an unprecedented sensitivity, in order to confirm or refute the DAMA/LIBRA claim [1]. The ultimate goal of SABRE is to deploy two independent NaI(Tl) crystal arrays in the northern (SABRE North) and southern (SABRE South) hemispheres to identify possible contributions to the modulation from seasonal or site-related effects. The two detectors have various common features as they use the same detector module concept, simulation, DAQ, and software frameworks, but they differ in their shielding designs: SABRE North, unlike SABRE South, will not use a liquid scintillator veto. For information on the SABRE South set up, see Ref. [2].

Table 1: Activities of background components in other NaI(Tl)-based experiments [1,3–5]. Secular equilibrium is assumed for both $^{238}$U and $^{232}$Th by all experiments. The range of values for $^{210}$Pb (PTFE) in the COSINE experiment is derived from Fig.8 of [4] by multiplying for the mass of the crystals.

| Source | Activity in DAMA/LIBRA crystals [mBq/kg] | Activity in ANAIS crystals [mBq/kg] | Activity in COSINE crystal [mBq/kg] |
|---|---|---|---|
| $^{40}$K | ≤0.62 | 0.70-1.33 | 0.58-2.5 |
| $^{210}$Pb (bulk) | 0.005-0.03 | 0.70-3.15 | 0.74-3.20 |
| $^{238}$U | 0.009-0.12 | 0.003-0.01 | 0.0002-0.001 |
| $^{232}$Th | 0.002-0.03 | 0.0004-0.004 | 0.001-0.01 |
| $^{3}$H | ≤0.09 | 0.09-0.20 | 0.05-0.12 |
| $^{129}$I | 0.96±0.06 | 0.96±0.06 | 0.72-1.08 |
| $^{210}$Pb (PTFE) | - | 0-3 mBq | 0-8 mBq |

SABRE aims to a background rate of 0.3-0.5 counts/day/kg/keV (cpd/kg/keV) in the 1-6 keV energy region of interest (ROI) for DM search, to achieve the ultimate verification of the DAMA result in about three years of data-taking and with a total mass of just a fraction of the present generation experiments. The key element in achieving this ambitious goal is the use of ultra-high purity NaI(Tl) crystals. Indeed, as already demonstrated by other existing NaI(Tl)-based experiments, such as DAMA/LIBRA, ANAIS-112 and COSINE-100 [1, 3–5], a large fraction of the background in the ROI for DM search comes from residual radioactive contaminants in the crystal themselves, especially $^{40}$K and $^{210}$Pb (Tab. 1).

In this pathway, the crystal denominated NaI-33 was thoroughly characterized inside both the SABRE PoP detector [6] in 2020 and the SABRE PoP-dry detector [7] in 2021 at Laboratori Nazionali del Gran Sasso (LNGS).

# 2 Experimental setups

We characterized the 3.4-kg crystal NaI-33 by operating two different detectors at LNGS. The detailed description of the two setups can be found in Ref. [8] and [7], but we summarized the most important features in the following subsections (2.1 and 2.2).

## 2.1 The PoP detector

The SABRE PoP detector was commissioned between May and July 2020, and took data until September 2020, accumulating an exposure of 90 kg×days. Its goal was to assess the radio-purity of SABRE crystals and test the performance of an external veto for the suppression of $^{40}$K and other gamma-emitting background from internal contaminations of the crystal. In that setup, a detector module consisting of a NaI(Tl) crystal wrapped with PTFE reflector and directly coupled to two photomultiplier tubes (PMTs), was placed inside a stainless steel vessel containing a 2-ton liquid scintillator veto. A passive shielding made of polyethylene (∼40 cm) and water (80-90 cm) surrounded the whole apparatus to further reduce external backgrounds, especially gammas from radioactive decays in the laboratory environment.

## 2.2 The PoP-dry detector

The PoP-dry setup was commissioned in March 2021 and stems from the SABRE-PoP. After removing the veto vessel, together with the liquid scintillator, we placed the NaI-33 detector module directly inside the PoP passive shielding (polyethylene plus water). To compensate for

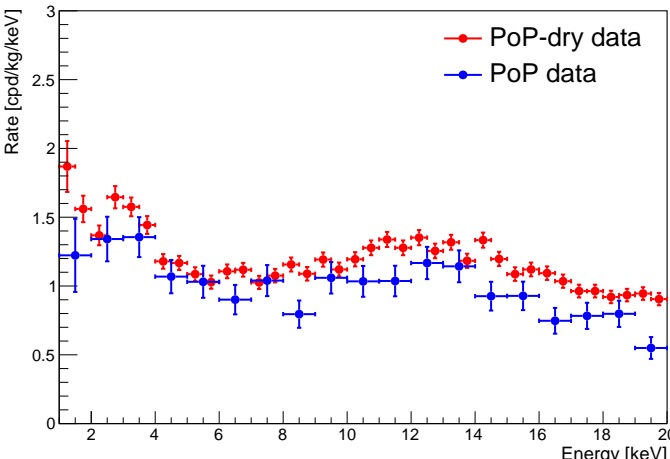

Figure 1: NaI-33 low energy spectrum (after selection cuts and acceptance-correction) for data acquired within the PoP-dry (red points) and the PoP setup (blue points). The latter includes only events in anti-coincidence with the liquid scintillator veto. The wider binning (and error bars) reflects the ∼10 times lower exposure.

the missing shielding power of the liquid scintillator (∼70 cm), we added a low radioactivity copper layer (10 cm on all sides and top, 15 cm below) and some additional polyethylene slabs around the copper. The inner volume of the detector module and of the shielding are continuously flushed with high-purity nitrogen gas to avoid moisture and radon.

The data analysis reported here refers to the data acquired between March 17, 2021 and February 25, 2022, for a total exposure of 891 kg×days.

## 3 Data analysis and results

The process of building the energy spectrum in a NaI(Tl) detector requires the selection of events to be retained as scintillation signal and those to be discarded as noise. Our selection is based on a set of variables, defined in detail in Ref. [9].

For the PoP data analysis we applied a traditional cut-based selection as described in [6], while for the PoP-dry we chose to exploit a multivariate approach based on Boosted Decision Trees (BDT) [10], in order to maximize the signal acceptance at very low energies [6].

Fig. 1 shows the NaI-33 energy spectrum below 20 keV acquired with the PoP-dry setup (red points), along with the one acquired with the PoP setup (blue points) and which benefits of the anti-coincidence with the liquid scintillator veto. Both the energy spectra are acceptance-corrected according to the applied event selection criterion. The resulting average count rate in the 1-6 keV ROI obtained within the PoP-dry is 1.39 ± 0.02 cpd/kg/keV [7]. This background level is comparable to that measured within the PoP setup, i.e. 1.20 ± 0.05 cpd/kg/keV [6]. Such result demonstrates that, as the vetoable crystal internal contaminants (e.g. $^{40}$K) are low enough, the active veto is no longer necessary to achieve the SABRE background goal.

In both cases we performed a fit of the energy spectrum with a combination of several background components. The spectral shapes of such components are calculated via our Monte Carlo [11] simulation code. In particular, we included: $^{40}$K, $^{210}$Pb, $^{3}$H, $^{226}$Ra, $^{232}$Th, $^{129}$I, and a flat component which accounts for $^{87}$Rb and other internal and external contributions (such as radioactivity in PMTs). A specific $^{238}$U contribution from the PMTs quartz window was included in the fit of PoP-dry data to better reproduce the experimental energy spectrum around 16 keV. In addition, $^{210}$Pb from the PTFE reflector wrapping the crystal was included

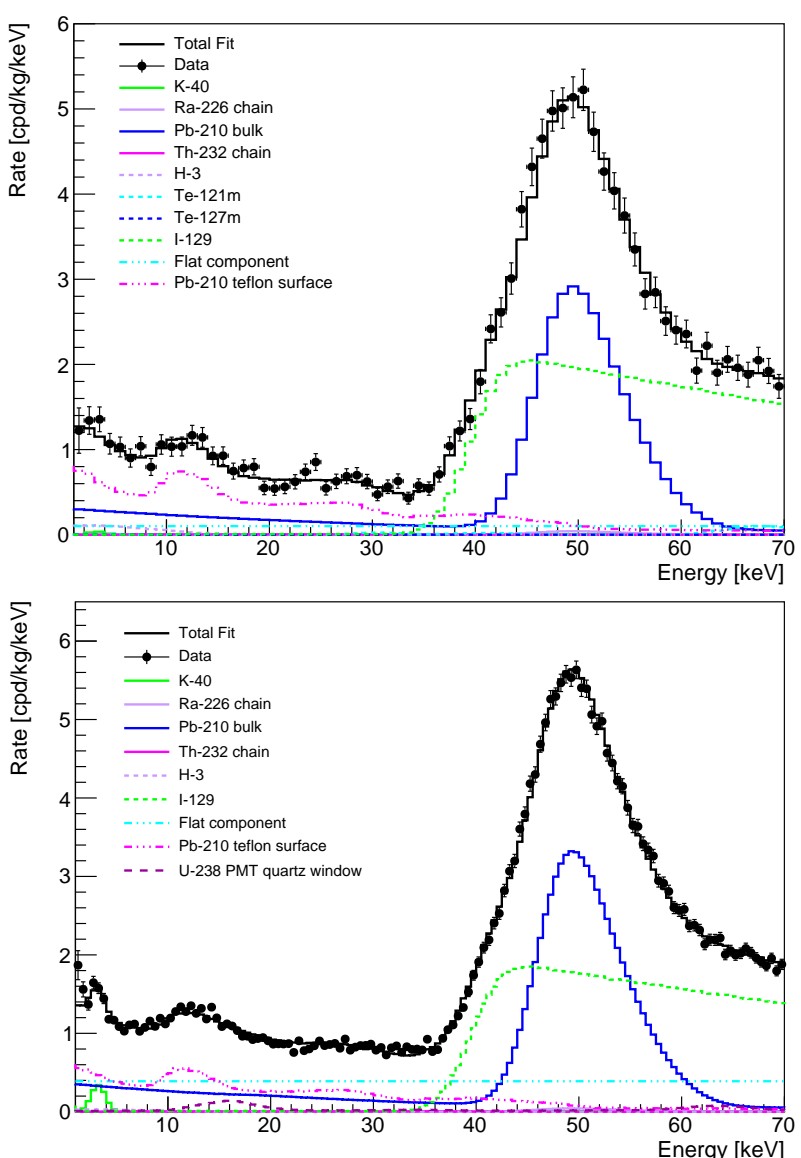

Figure 2: PoP (left) and PoP-dry (right) energy spectrum of the NaI-33 crystal up to 70 keV with a spectral fit. Data are shown after noise rejection and acceptance correction.

to reproduce the peak at $\sim$12 keV due to X-rays from $^{210}$Pb.

The result of the best-fit in the 2-70 keV energy range is shown in Fig. 2 (by extrapolating the spectrum behaviour below 2 keV) for both PoP ($\chi^2/N_{dof} = 96/88$) and PoP-dry ($\chi^2/N_{d.o.f.} = 177/127$) data. Tab. 2 summarises the activities of the different background components determined from the spectral fits (2nd and 4th columns) and the corresponding rate in the 1-6 keV ROI (3rd and 5th columns). The activities of the internal background sources are consistent between the two analyses. The dominant background contributions come from a $^{210}$Pb contamination in the PTFE reflector and in the crystal bulk. It should be noted that the PoP-dry passive shielding is not yet optimized for an high-sensitivity full-scale experiment. Consequently also the flat component gives a significant contribution in the ROI (probably due to environmental gammas entering the shielding). The $^{40}$K contribution instead, is sub-dominant also in the PoP-dry, due to the ultra-high purity of NaI-33 in terms of potassium.

Table 2: Activities and current rate in ROI (1-6 keV) of different background components in NaI-33 from the spectral fit of PoP and PoP-dry data. The activities of $^{210}$Pb in the PTFE reflector and of $^{238}$U in the PMTS quartz window are given in mBq. Upper limits are given as one-sided 90% CL. The total rates are conservatively calculated using upper limits.

| | PoP | | PoP-dry | |
|---|---|---|---|---|
| **Source** | **Activity in NaI-33 [mBq/kg]** | **Rate in ROI in NaI-33 [cpd/kg/keV]** | **Activity in NaI-33 [mBq/kg]** | **Rate in ROI in NaI-33 [cpd/kg/keV]** |
| $^{40}$K | 0.14±0.01 | 0.02±0.01 | 0.15±0.02 | 0.12±0.02 |
| $^{210}$Pb (bulk) | 0.42±0.02 | 0.28±0.01 | 0.46±0.01 | 0.32±0.04 |
| $^{226}$Ra | 0.006±0.001 | 0.004±0.001 | 0.006±0.001 | 0.004±0.001 |
| $^{232}$Th | 0.002±0.001 | | 0.002±0.001 | |
| $^{3}$H | 0.012±0.007 | ≤0.13 | ≤0.005 | ≤0.05 |
| $^{129}$I | 1.34±0.04 | | 1.29±0.02 | |
| $^{210}$Pb (PTFE) | 1.01±0.20 mBq | 0.63±0.09 | 0.83±0.06 mBq | 0.46±0.03 |
| $^{238}$U (PMT quartz window) | - | - | 0.31±0.05 mBq | 0.011±0.002 |
| Other (flat) | | 0.10±0.05 | | 0.39±0.02 |
| **total** | | 1.16±0.10 | | 1.36±0.04 |

# 4 Conclusion

In this work, we described the characterization of the low background NaI(Tl) SABRE crystal NaI-33 into two different setups at LNGS: the PoP, equipped with a liquid scintillator veto, and the PoP-dry, a modified SABRE PoP setup that does not feature the use of such veto. The PoP-dry data show a count rate in the 1-6 keV ROI of 1.39 ± 0.02 cpd/kg/keV: a minor increase with respect to the 1.20 ± 0.05 cpd/kg/keV observed in the PoP, and considering that the PoP-dry shielding leaves room for sizable improvement in the next future. We have built a background model fitting both energy spectra with a combination of several Monte Carlo simulated components, and we find consistent results between the PoP and PoP-dry data analysis. The dominant background is actually not affected by the veto and can be ascribed to $^{210}$Pb: this is present in the crystal bulk, but it mostly spawns from a significant $^{210}$Pb contamination in the PTFE reflector. This conclusion open to a new design of the experimental setup for the SABRE North physics phase. The detector will be based on an array of crystals with radio-purity similar to NaI-33, wrapped in a specially selected low radioactivity PTFE reflector and it will feature a further improved purely passive shielding.

# Acknowledgements

**Funding information** This work was supported by INFN funding and National Science Foundation under the Awards No. PHY-1242625, No. PHY-1506397, and No. PHY-1620085.

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
