# Peer review of "Status and prospects of SABRE North"

_SciPost Physics Proceedings, doi:SciPost Phys. Proc. 12, 026 (2023)_

## Round 1 · Referee Report · Anonymous (Referee 1) · 2022-11-6

Report

In this proceeding author describes the characterization of background of NaI doped by Tl crystal studies in 2 setups. The results of analysis and developed background model presented. The manuscript is clearly written and well organised. It is also suitably formatted for publication. I recommend the manuscript for publication after some minor comments: - I was missing comparison of activities of key background contributors with ones in DAMA experiment or others. - If your result proofs that liquid veto is not needed, what is the motivation to have one in SABRE South? - Was not clear to me, where this crystal will be used? North?

Requested changes

  • I was missing comparison of activities of key background contributors with ones in DAMA experiment or others.
  • If your result proofs that liquid veto is not needed, what is the motivation to have one in SABRE South?
  • Was not clear to me, where this crystal will be used? North?

  • validity: high
  • significance: high
  • originality: -
  • clarity: -
  • formatting: excellent
  • grammar: -

Author:  Ambra Mariani  on 2022-12-20  [id 3162]

(in reply to Report 1 on 2022-11-06)
Category:
answer to question

1) Added a table containing the activities of the main background components measured by other NaI-based experiments. The table is separated from that of NaI-33 for space reasons.

2) The SABRE North Collaboration demonstrated that the use of the liquid scintillator is not strictly necessary to verify the DAMA/LIBRA result with the present level of background. In particular, this is true for the NaI-33 crystal, due to its very low content of K-40. However, since for the SABRE South laboratory there are no restrictions regarding the use of a liquid scintillator veto, and also the environmental radioactivity in the laboratory has not yet been fully measured, they could benefit of the use of the liquid scintillator.

3) This crystal will be used in the SABRE North setup.

---

## Round 2 · List of Changes

Added a table containing the activities of the main background components of other NaI-based experiments.

You are currently on this page

Resubmission scipost_202210_00014v2 on 20 December 2022

---

## Editorial Decision

published